# Molecular and phenotypic characteristics of RSV infections in infants during two nirsevimab randomized clinical trials

Bahar Ahani[1,11], Kevin M. Tuffy[2,11], Anastasia A. Aksyuk [2], Deidre Wilkins[2], Michael E. Abram [2], Ron Dagan [3], Joseph B. Domachowske[4], Johnathan D. Guest [5], Hong Ji[2], Anna Kushnir[2], Amanda Leach[6], Shabir A. Madhi [7], Vaishali S. Mankad[8], Eric A. F. Simões[9], Benjamin Sparklin [1], Scott D. Speer[2], Ann Marie Stanley [2], David E. Tabor[2], Ulrika Wählby Hamrén[10], Elizabeth J. Kelly [2] ✉ & Tonya Villafana[6]

Nirsevimab is a monoclonal antibody that binds to the respiratory syncytial virus (RSV) fusion protein. During the Phase 2b (NCT02878330) and MELODY (NCT03979313) clinical trials, infants received one dose of nirsevimab or placebo before their first RSV season. In this pre-specified analysis, isolates from RSV infections were subtyped, sequenced and analyzed for nirsevimab binding site substitutions; subsequently, recombinant RSVs were engineered for microneutralization susceptibility testing. Here we show that the frequency of infections caused by subtypes A and B is similar across and within the two trials. In addition, RSV A had one and RSV B had 10 fusion protein substitutions occurring at >5% frequency. Notably, RSV B binding site substitutions were rare, except for the highly prevalent I206M:Q209R, which increases nirsevimab susceptibility; RSV B isolates from two participants had binding site substitutions that reduce nirsevimab susceptibility. Overall, >99% of isolates from the Phase 2b and MELODY trials retained susceptibility to nirsevimab.

Respiratory syncytial virus (RSV) is a major cause of lower respiratory tract infection (LRTI) and hospitalization in infants and young children globally[1]. Severe RSV disease occurs primarily in infants younger than 6 months of age[2] and most hospitalizations for RSV LRTI (66–79%) occur in otherwise-healthy infants born at term[3–6].

Nirsevimab is a recombinant human immunoglobulin G1 kappa monoclonal antibody with an extended half-life[7]. It binds the F1 and F2 subunits of the RSV fusion (F) protein at the highly conserved antigen site Ø, locking the RSV F protein in the pre-fusion conformation, thereby blocking viral entry into the host cell[7,8]. Nirsevimab is being developed for the general infant population, including infants in their first year of life and those remaining medically vulnerable in their second year of life. In a Phase 2b placebo-controlled randomized clinical trial (NCT02878330), a single 50 mg intramuscular (IM) dose of

[1]Bioinformatics, Vaccines & Immune Therapies, BioPharmaceuticals R&D, AstraZeneca, Gaithersburg, MD, USA. [2]Translational Medicine, Vaccines & Immune Therapies, BioPharmaceuticals R&D, AstraZeneca, Gaithersburg, MD, USA. [3]The Shraga Segal Department of Microbiology, Immunology and Genetics, Faculty of Health Sciences of the Ben-Gurion University of the Negev, Beer-Sheva, Israel. [4]State University of New York Upstate Medical University, Syracuse, NY, USA. [5]Virology and Vaccine Discovery, Vaccines & Immune Therapies, BioPharmaceuticals R&D, AstraZeneca, Gaithersburg, MD, USA. [6]Clinical Development, Vaccines & Immune Therapies, BioPharmaceuticals R&D, AstraZeneca, Gaithersburg, MD, USA. [7]South African Medical Research Council Vaccines and Infectious Diseases Analytics Research Unit, Faculty of Health Sciences, University of the Witwatersrand, Johannesburg, South Africa. [8]Clinical Development, Vaccines & Immune Therapies, BioPharmaceuticals R&D, AstraZeneca, Durham, NC, USA. [9]University of Colorado School of Medicine and Children's Hospital Colorado, Aurora, CO, USA. [10]Clinical Pharmacology and Quantitative Pharmacology, R&D, AstraZeneca, Gothenburg, Sweden. [11]These authors contributed equally: Bahar Ahani, Kevin M. Tuffy. ✉e-mail: beth.kelly@astrazeneca.com

nirsevimab administered before the RSV season was 70.1% effective at preventing medically attended (MA) RSV LRTI among healthy pre-term infants born at a gestational age of 29 to <35 weeks over 150 days post-dose, equivalent to an RSV season[9]. Based on pharmacokinetic (PK) and drug exposure–response analyses from this trial, the fixed-dose strategy was optimized to a weight-banded regimen, where participants <5 kg received 50 mg and those ≥5 kg received 100 mg nirsevimab[10]. The weight-banded regimen was subsequently used for all studies going forward, including the Phase 3 MELODY trial (NCT03979313) in late pre-term and term (gestational age of ≥35 weeks) infants[11]. In a pooled analysis of infants receiving the weight-banded dosing in Phase 2b trial and infants in the MELODY trial, nirsevimab demonstrated an efficacy of 79.5% against MA RSV LRTI and 77.3% against hospitalization for RSV LRTI over 150 days post-dose[10]. Notably, recent data demonstrate that nirsevimab protects from RSV disease without inhibiting an adaptive immune response[12].

The Phase 3 clinical trial of a previous investigational monoclonal antibody developed to prevent MA RSV LRTI in pre-term infants (suptavumab) failed to meet its primary endpoint because of the emergence of a newly circulating strain of RSV B with amino acid substitutions in the F protein that prevented binding to and neutralization of RSV[13]. As suptavumab attaches through antigen site V rather than site Ø, this substitution would not affect nirsevimab binding; however, it clearly demonstrates the importance and need for ongoing genotypic evaluation of RSV.

The aims of the present study were to genotypically evaluate RSV infections occurring in two pivotal nirsevimab clinical trials (Phase 2b: NCT02878330; MELODY: NCT03979313), characterize substitutions in the nirsevimab binding site, and determine their effects on nirsevimab susceptibility of RSV A and B isolates identified as causing infections. The prevalence of substitutions observed during these clinical trials was also evaluated in relation to changes observed across circulating RSV strains from surveillance data.

## Results

### Study population

Together, the Phase 2b and MELODY trials enrolled infants across 4 years in both the Northern and Southern Hemispheres in both inpatient and outpatient settings. The Phase 2b study was performed at 164 sites in 23 countries; MELODY was performed at 160 sites in 21 countries. Of 1453 participants randomized in Phase 2b trial, next-generation sequencing (NGS)-evaluable RSV isolates were identified in 40 of 969 nirsevimab recipients and 54 of 484 placebo recipients who had suspected LRTI or any respiratory illness requiring hospitalization over 360 days post-dose. Of those who received nirsevimab, 17 received 50 mg nirsevimab and were <5 kg in body weight at the time of injection, in line with the subsequent weight-banded dosing regimen, whereas 23 participants received 50 mg nirsevimab but were ≥5 kg and thus received a dose that was later determined to result in suboptimal exposures to nirsevimab (Supplementary Fig. S1A). In the full MELODY cohort of 3012 randomized participants (including the initial 1490 participants randomized for the primary analysis), NGS-evaluable RSV was identified in 60 of 2009 nirsevimab recipients and 88 of 1003 placebo recipients with suspected LRTI over 510 days post-dose who met any case definition (Supplementary Fig. S1B). Data through Day 510 post-dose were only available for the MELODY primary cohort at the time of this analysis.

### Genotypic characterization of RSV isolates

RSV A and B subtypes were detected with similar frequencies in infants with MA RSV LRTI (Fig. 1A) and those with RSV LRTI requiring hospitalization (Fig. 1B). A higher total number of RSV cases

**Placebo** — RSV A, RSV B **Nirsevimab** — RSV A, RSV B

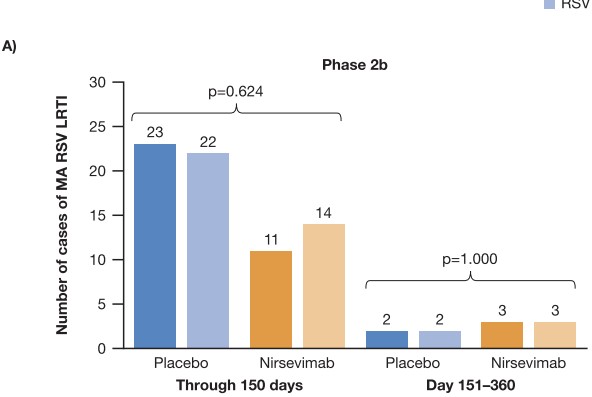

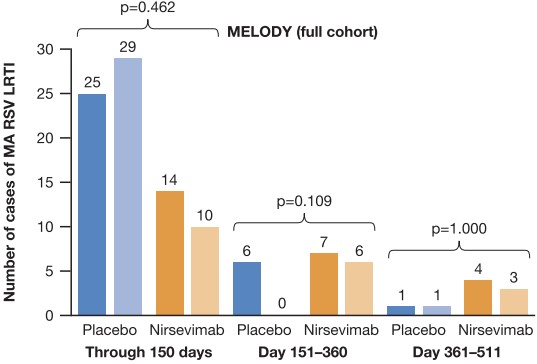

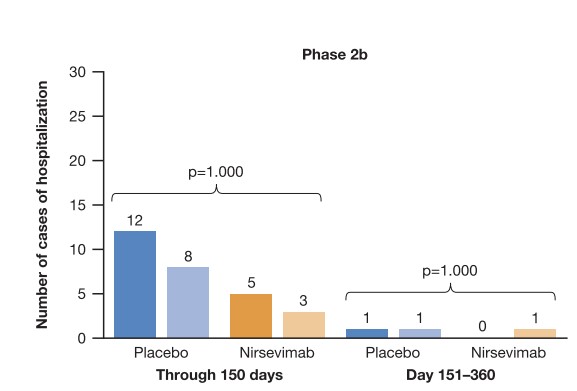

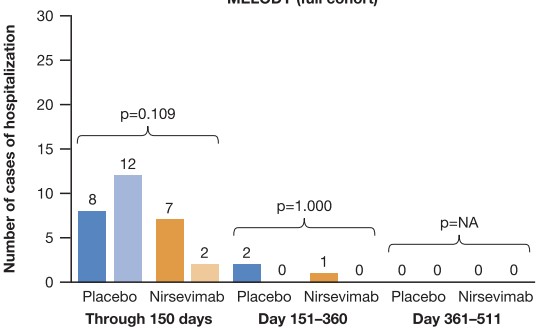

**Fig. 1 | Incidence of RSV A and RSV B subtypes in the Phase 2b and MELODY trials (full cohort). A** MA RSV LRTI and **B** hospitalization due to MA RSV LRTI. Cases shown are those with evaluable next-generation sequencing data. Two-sided Fisher's exact tests comparing cases of RSV A and RSV B found no statistical difference by endpoint or timepoint. LRTI lower respiratory tract infection, MA medically attended, RSV respiratory syncytial virus.

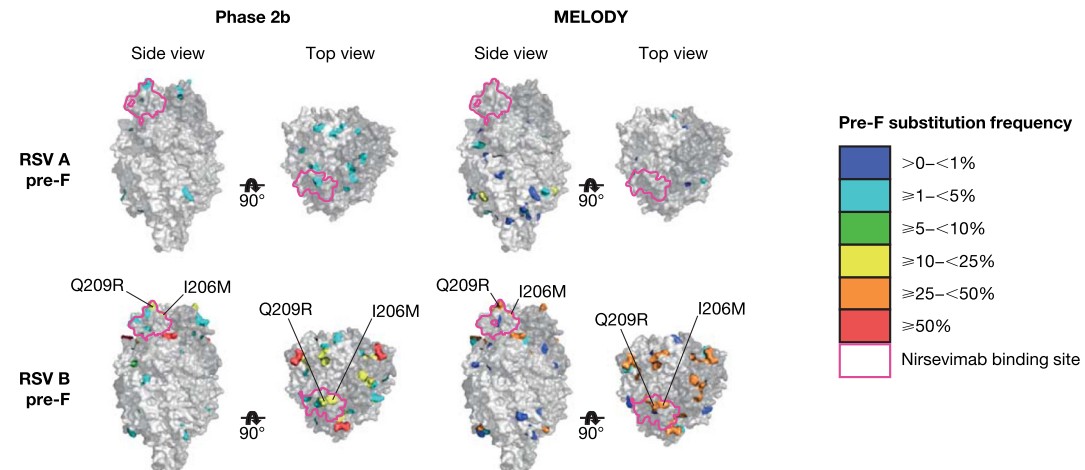

**Fig. 2 | Major observed pre-F substitutions in the Phase 2b and MELODY trials (full cohort). A** Location and **B** frequency. RSV A residues S25, S99, A102, and A103 with substitutions frequencies ≥1–<5% from Phase 2b; S24, S105, and A518 with substitutions frequencies >0–<1% from MELODY; and A103, A107 with substitutions frequencies ≥1–<5% from MELODY are not depicted due to absence or density in PDB ID: 5UDC. RSV B residues A103 with substitutions frequency ≥50% from Phase 2b; T518 with substitutions frequency ≥1–<5% from Phase 2b; N99, and T522 with substitutions frequencies >0–<1% from MELODY; and A103V with substitutions frequency ≥25– < 50% from MELODY are not depicted due to absence or density in PDB ID: 5UDD. F fusion protein, PDB Protein Data Bank, RSV respiratory syncytial virus.

occurred among placebo recipients than nirsevimab recipients, despite the 2:1 randomization of nirsevimab to placebo. Most cases of MA RSV LRTI occurred during the first 150 days post-dose, which corresponds to the average length of an RSV season.

Few major variant amino acid substitutions (molecular allele frequency [MAF] ≥25%) were observed in either RSV A or B subtypes within 150 days post-dose and most of those were present in the non-extracellular (EC) region of the F protein (Supplementary Tables S2–S6). Of RSV A isolates with major variant substitutions, one non-binding EC substitution occurred with a frequency >5% in MELODY (I384T; 11.5%). A single binding site substitution was identified in RSV A with a frequency <5% (K209R; 2.1%), occurring in isolates collected from two nirsevimab recipients in Phase 2b trial; both participants had illnesses that met the exploratory endpoint definition for RSV infection (RSV unscheduled event). In contrast, there were 10 unique major variant substitutions in RSV B that occurred with a frequency >5% in the Phase 2b and MELODY trials, two of which (I206M, Q209R) were in the binding site region (Figs. 2 and 3). These binding site substitutions were identified in both placebo and nirsevimab recipients, consistent with their high circulation in recent surveillance studies[14]. L204S and S211N binding site substitutions were observed during the MELODY trial with a frequency <5%; both co-occurred with

I206M:Q209R. Three additional substitutions in the RSV B binding site were observed in two participants in Phase 2b trial: I64T:K68E in one participant (co-occurring with I206M:Q209R) and N208S in a separate participant[14].

Substitutions in the palivizumab binding site were rarely identified in the Phase 2b and MELODY trials. In the MELODY trial, one participant who met the primary case definition and received nirsevimab had a major variant K272R substitution in antigenic site II prior to Day 150 post-dose.

### Phenotypic characterization of observed RSV F substitutions

The neutralization potency of nirsevimab was retained for all RSV isolates containing binding site changes identified from participants in the weight-banded dosing regimen with MA RSV LRTI (Tables 1, 2 and Supplementary Tables S7–S12). Only three substitutions (I64T, N208S, and K68E) from two nirsevimab recipients infected with RSV B (both in the Phase 2b trial) were associated with nirsevimab resistance, with increases of >200-fold in the half-maximal inhibitory concentration ($IC_{50}$) (Fig. 4). These isolates were collected from two participants that both weighed ≥5 kg and had received the 50 mg dose (later determined to be suboptimal), but also had serum levels of nirsevimab within the range of Phase 2b participants weighing <5 kg

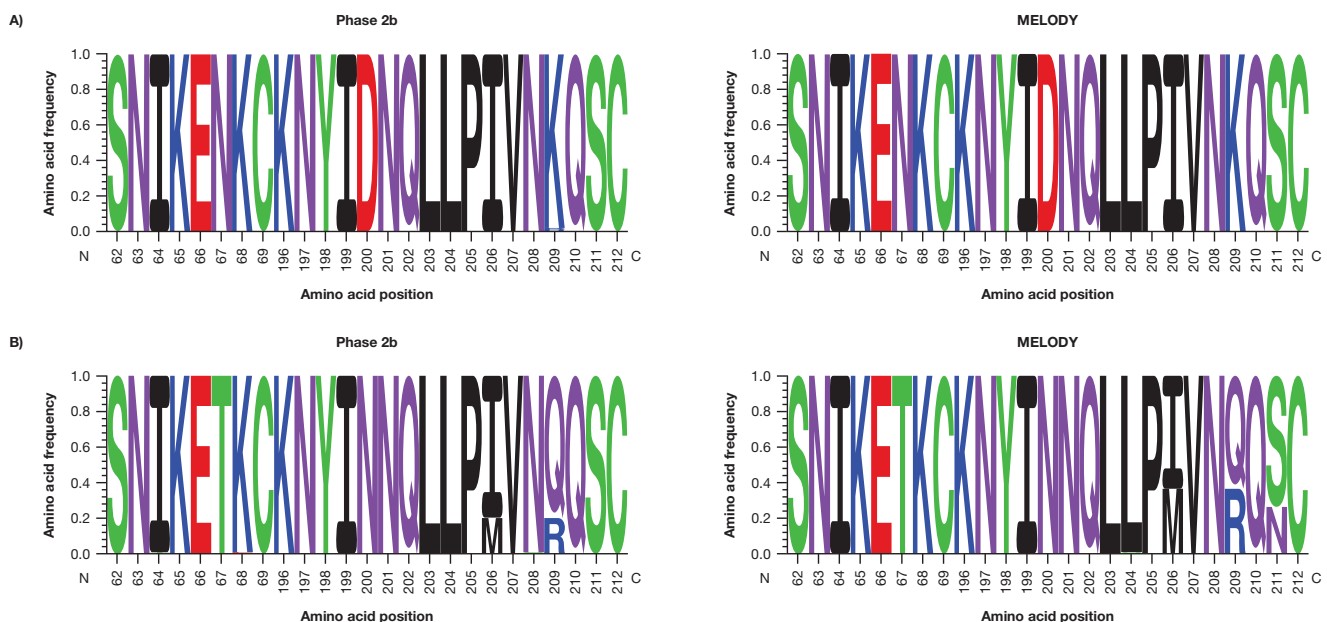

**Fig. 3 | Frequency of major variant substitutions in the nirsevimab binding site in the Phase 2b and MELODY trials (full cohort). A** RSV A and **B** RSV B. Weblogos (i.e., Sequence Logos) were generated in Seq2Logo-2.0 (https://services.healthtech. dtu.dk/service.php?Seq2Logo-2.0)[24]. Amino acids are differentiated by color: green, polar; purple, neutral; blue, basic; and red, acidic. RSV respiratory syncytial virus.

**Table 1 | Genotype and phenotype of major variant substitutions in the F protein of the nirsevimab binding site and changes to nirsevimab susceptibility in participants with confirmed MA RSV LRTI meeting the primary case definition in Phase 2b trial through 150 days post-dose**

| Amino acid substitutions | Prevalence in surveillance studies (%) | Placebo | Nirsevimab | Total | Fold reduction in susceptibility | | IC$_{50}$ (ng/mL) | |
|---|---|---|---|---|---|---|---|---|
| | | | | | Nirsevimab | Palivizumab | Nirsevimab | Palivizumab |
| **RSV A**[a] | | (n = 23) | (n = 11) | (n = 34) | | | | |
| **No change** | NA | 23 (100%) | 11 (100%) | 34 (100%) | 1.0 | 1.0 | NA | NA |
| **RSV B** | | (n = 22) | (n = 14) | (n = 36) | | | | |
| **No change** | NA | 14 (63.6%) | 5 (35.7%) | 19 (52.8%) | 1.0 | 1.0 | NA | NA |
| **I206M:Q209R** | 66 | 8 (36.4%) | 3 (21.4%) | 11 (30.6%) | 0.2 | 1.3 | 0.4 | 79.8 |
| **I206M**[b] | 68.89 | 8 (36.4%) | 4 (28.6%) | 12 (33.3%) | 5.0 | 2.0 | 7.0 | 164.4 |
| **Q209R**[c] | 68.18 | 8 (36.4%) | 4 (28.6%) | 12 (33.3%) | 0.5 | 3.1 | 0.9 | 213.7 |
| **I64T:K68E: I206M:Q209R** | NA | 0 | 1 (7.1%) | 1 (2.8%) | >447.1 | 1.2 | >600 | 122.0 |
| **I64T**[d] | NA | 0 | 1 (7.1%)[d] | 1 (2.8%) | >496.3 | 5.2 | >600[d] | 454.0 |
| **K68E**[e] | NA | 0 | 1 (7.1%) | 1 (2.8%) | >283.4 | 2.1 | 8.5 | 120.8 |
| **N208S** | NA | 0 | 1 (7.1%) | 1 (2.8%) | >386.6 | 1.8 | >600 | 84.1 |

% of participants with RSV containing substitutions in the nirsevimab binding site is calculated based on the total number of participants meeting the primary endpoint of MA RSV LRTI in each treatment arm (placebo, nirsevimab) and each RSV strain (RSV A, RSV B).

*LRTI* lower respiratory tract infection, *MA* medically attended, *NA* not available, *NGS* next-generation sequencing, *RSV* respiratory syncytial virus.

[a]Two infants in the Phase 2b trial with RSV A who met the primary case definition through Day 150 as previously published in ref. 9 were not included in the analysis due to having no evaluable NGS sequencing data.

[b]Observed only as co-occurring with Q209R.

[c]Observed only as co-occurring with I206M.

[d]Observed only as co-occurring with K68E:I206M:Q209R.

[e]Observed only as co-occurring with I64T:I206M:Q209R.

(Fig. 5; see Supplementary Note 1 for case narratives). All substitutions retained susceptibility to palivizumab; for the palivizumab binding site substitution K272R identified in MELODY, a moderate shift in susceptibility to palivizumab (fold change in IC$_{50}$ = 41.8) was noted. Of note, no resistance-associated substitutions in the nirsevimab binding site were identified beyond 150 days post-dose.

Susceptibility to nirsevimab increased for RSV B isolates with the common I206M:Q209R substitution, as well as for additional substitutions co-occurring with I206M:Q209R (Tables 1, 2 and Fig. 6). The

S211N substitutions noted in the RSV binding site, alone or with observed co-occurring substitutions, did not reduce the neutralization potency of nirsevimab. Data on L204S substitutions (either engineered as a single substitution or with observed co-occurring substitutions I206M:Q209R:S211N) were unavailable at the time of this analysis. Given the importance of antigenic site Ø as a target of neutralizing antibodies in the immune response to RSV, resistance-associated substitutions identified in clinical studies were evaluated for their ability to be neutralized by serum from 17 healthy donors

**Table 2 | Genotype and phenotype of major variant substitutions in the F protein of the nirsevimab binding site and changes to nirsevimab susceptibility in participants with confirmed MA RSV LRTI meeting the primary case definition in MELODY through 150 days post-dose**

| Amino acid substitutions | Prevalence in surveillance studies (%) | Placebo | Nirsevimab | Total | Fold reduction in susceptibility | | IC$_{50}$ (ng/mL) | |
|---|---|---|---|---|---|---|---|---|
| | | | | | Nirsevimab | Palivizumab | Nirsevimab | Palivizumab |
| **RSV A** | | (n = 23) | (n = 13) | (n = 36) | | | | |
| No change | NA | 23 (100%) | 13 (100%) | 36 (100%) | 1.0 | 1.0 | NA | NA |
| **RSV B** | | (n = 29) | (n = 10) | (n = 39) | | | | |
| No change | NA | 1 (3.4%) | 0 | 1 (2.6%) | 1.0 | 1.0 | NA | NA |
| I206M:Q209R | 66.0 | 6 (20.7%) | 1 (10%) | 7 (17.9%) | 0.2 | 1.3 | 0.4 | 79.8 |
| I206M[a] | 68.9 | 28 (96.6%) | 10 (100%) | 38 (97.4%) | 5 | 2.0 | 7.0 | 164.4 |
| Q209R[b] | 68.2 | 28 (96.6%) | 10 (100%) | 38 (97.4%) | 0.5 | 3.1 | 0.9 | 213.7 |
| I206M:Q209R:S211N | 1.1 | 22 (75.9%) | 8 (80%) | 30 (76.9%) | 0.5 | 3.7 | 0.9 | 204.4 |
| S211N[c] | 1.1 | 22 (75.9%) | 9 (90%) | 31 (79.5%) | 1.2 | 1.9 | 2.5 | 148.7 |
| L204S:I206M:Q209R:S211N | 0 | | 1 (10%) | 1 (2.6%) | ND | ND | ND | ND |
| L204S[d] | 0 | | 1 (10%) | 1 (2.6%) | ND | ND | ND | ND |

% of participants with RSV containing substitutions in the nirsevimab binding site is calculated based on the total number of participants meeting the primary endpoint of MA RSV LRTI in each treatment arm (placebo, nirsevimab) and each RSV strain (RSV A, RSV B).

*LRTI* lower respiratory tract infection, *MA* medically attended, *NA* not available, *ND* not determined, *NGS* next-generation sequencing, *RSV* respiratory syncytial virus.

[a]Observed as co-occurring with Q209R ± L204S, S211N.
[b]Observed as co-occurring with I206M ± L204S, S211N.
[c]Observed only as co-occurring with I206M:Q209R.
[d]Observed only as co-occurring with I206M:Q209R:S211N.

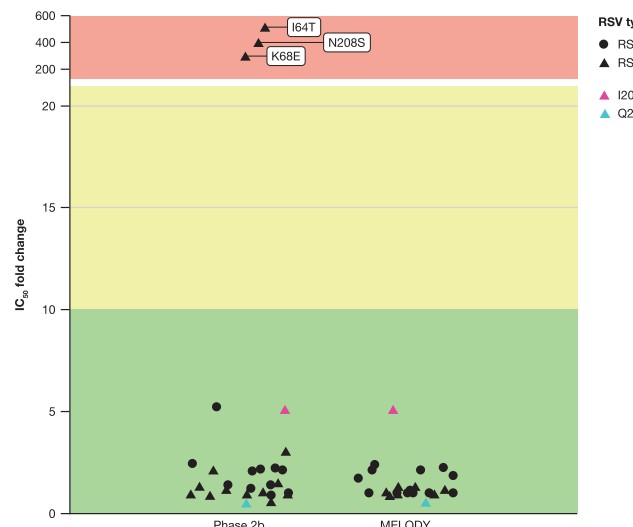

**Fig. 4 | Change in nirsevimab IC50 for major RSV F protein variants evaluated in the Phase 2b and MELODY trials (full cohort).** Individual major variant substitutions from Phase 2b and MELODY trials were evaluated in a validated RSV neutralization susceptibility assay (Viroclinics Biosciences BV, Rotterdam, NLD) and compared with RSV A and B reference viruses. Both I206M and Q209R were identified in RSV B. Green, yellow and red banding represents a low, medium, and high degree of IC$_{50}$ fold change, respectively. IC$_{50}$ half-maximal inhibitory concentration, RSV respiratory syncytial virus.

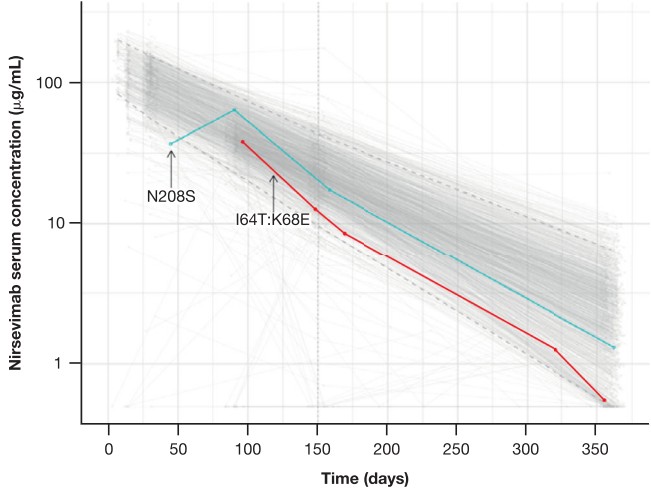

**Fig. 5 | Nirsevimab serum concentrations versus time after dose.** The blue and red lines/dots represent the nirsevimab serum concentrations from two participants with resistance-associated substitutions N208S and I64T:K68E who both weighed ≥5 kg and received the 50 mg dose in the Phase 2b trial. The gray lines/dots denote individual serum concentration-time profiles from participants in the Phase 2b trial <5 kg and MELODY (primary cohort) who had received weight-banded dosing; the broken lines are the 5th and 95th percentiles of the data. Samples below the limit of quantification (0.5 μg/mL) are shown as 0.5 μg/mL.

(Supplementary Fig. S2). Neutralization potency was found to be either similar or improved for both N208S and I64T:K68E substitutions compared with the reference virus.

At the time of symptomatic illness, RSV titers were similar in participants with breakthrough infections, regardless of whether the virus had resistance-associated substitutions (Supplementary Fig. S3). Median cycle threshold (CT) values were numerically higher (corresponding to lower viral load) among nirsevimab recipients compared

with placebo recipients in the Phase 2b trial, but similar in the MELODY trial, with no statistical significance shown in either trial.

**Molecular modeling**

The potential mechanism by which RSV B I206M:Q209R substitutions increased nirsevimab potency in vitro was assessed using molecular modeling. The I206M substitution was not implicated in direct interaction with nirsevimab, and despite the extended methionine side chain, the substitution was structurally tolerated (Fig. 7). The Q209

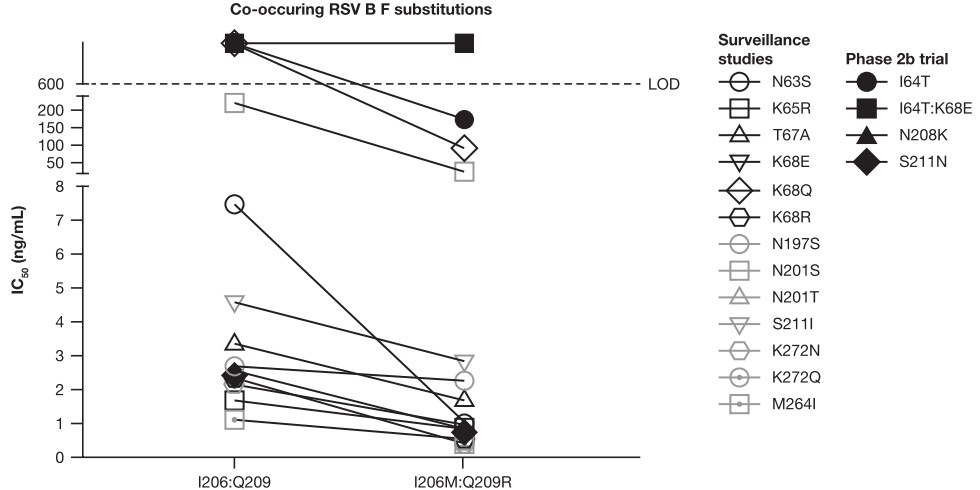

**Fig. 6 | Susceptibility of RSV B strains with and without prevalent nirsevimab binding site substitutions I206M + Q209R.** F, fusion protein; $IC_{50}$ half-maximal inhibitory concentration, LOD limit of detection, RSV respiratory syncytial virus.

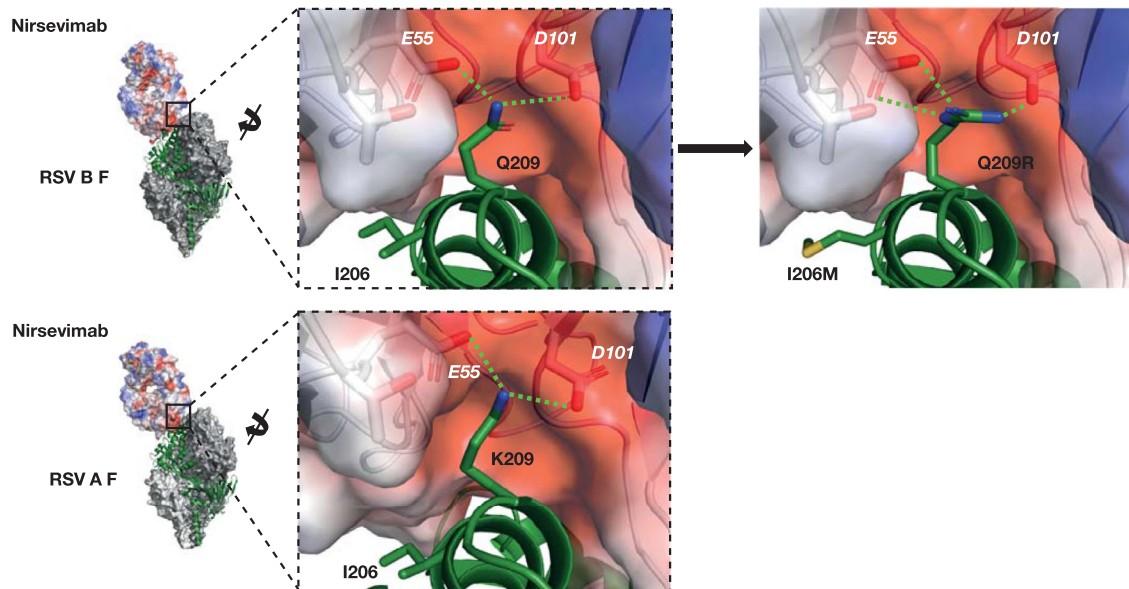

**Fig. 7 | Proposed mechanism of increased RSV B neutralization potency with nirsevimab binding site substitution Q209R.** Red indicates the electrostatics of the negatively charged pocket on nirsevimab, created by glutamic and aspartic acids (E55 and D101). Green dashed lines indicate polar contacts. F fusion protein, RSV respiratory syncytial virus.

residue is a known contact residue of nirsevimab[8] and its side chain occupies a negatively charged pocket on the antibody, created by the glutamic acid (E) and aspartic acid (D) residues at positions 55 (light chain) and 101 (heavy chain), respectively. The Q209R substitution results in the replacement of the polar glutamine (Q) side chain with a positively charged arginine (R) side chain, leading to improved electrostatic interactions with the E55 and D101 nirsevimab residues (Fig. 7). This conservative substitution is closely related to the positively charged lysine (K) at position 209 in the RSV A F protein, suggesting that both subtypes are interacting with the E55 and D101 nirsevimab pocket in a similar fashion. Further analysis of RSV F structures revealed substantial conformational changes in the nirsevimab epitope between unbound and antibody-bound states, with residue 209 shifting more than 7 Å in both subtypes (Supplementary Fig. S4) to accommodate these interactions. Together these data support a mechanism whereby the Q209R substitution in RSV B F improves critical contacts for antibody-antigen recognition.

A subsequent investigation into the RSV B I64T:K68E and N208S substitutions that were associated with nirsevimab resistance was also performed. I64, K68, and N208 have all been reported to interact with nirsevimab[8,15,16], and the resulting K68E and N208S in silico substitutions led to the loss and reduction of polar contacts, respectively (Supplementary Fig. S5). I64 is near the heavy chain CDR3 of nirsevimab and forms a close contact with V99 (<4 Å). However, the I64T in silico substitution did not impact the distance between these residues (Supplementary Fig. S5). Instead, the decrease in nirsevimab potency due to this substitution may stem from introducing a polar side chain into the site Ø-nirsevimab interface, thus disrupting a patch of hydrophobic interactions.

## Discussion
This study of infants with confirmed RSV, including LRTI and respiratory illnesses requiring hospitalization, demonstrated that nirsevimab effectively neutralized both RSV A and B infections among participants

enrolled in the Phase 2b and MELODY trials, regardless of the presence of naturally occurring amino acid substitutions, and including RSV infections that did not meet protocol defined endpoints. Genotypic analyses showed that most substitutions occurred at a low frequency and in non-EC portions of RSV F rather than in antigenic sites, indicating that the nirsevimab binding site remains conserved and thus RSV remains susceptible to nirsevimab.

No major RSV A binding site variant substitutions occurred in participants with MA RSV LRTI and hospitalization for RSV LRTI. However, two participants who met an exploratory RSV endpoint had RSV A that contained a binding site substitution (K209R) that did not affect nirsevimab susceptibility. While RSV B isolates had more binding site substitutions through 511 days post-dose, all but three substitutions identified in two participants were potently neutralized by nirsevimab. Additionally, of all the substitutions observed in RSV B in both placebo and nirsevimab recipients, only two occurred in the binding site at high frequency (I206M and Q209R) in the Phase 2b study, with one additional substitution, S211N, observed in the MELODY study. All these frequently observed binding site substitutions were effectively neutralized by nirsevimab. Recent surveillance studies have shown these substitutions to be highly prevalent in circulating RSV (28–>51% in 2021)[17] and are thus very unlikely to be treatment-associated. Prospective molecular surveillance studies have also shown that the nirsevimab binding site is highly conserved among circulating RSV strains (>98% since 2016) and resistance to nirsevimab has been rare and has not increased in geo-temporal frequency[14]. Of the isolates identified from two participants in the Phase 2b trial with resistance-associated substitutions (I64T, N208S, K68E), only N208S was identified in a viral escape analysis of nirsevimab[18]. However, when compared with the reference, the analysis did not demonstrate enhanced growth kinetics of the N208S variant in cell culture. Additionally, although a substitution at amino acid position 68 was described, it was a K68N substitution rather than the K68E substitution observed in the current study; this variant also displayed similar growth kinetics to the reference virus. Notably, these RSV variants were isolated following what was later determined to be suboptimal dosing; both participants had nirsevimab serum levels within the range of those in the weight-banded dosing regimen and also had similar viral titers to those of other nirsevimab recipients infected with RSV. These data align with recent reports that demonstrate no humoral antibody threshold is high enough to prevent mild RSV disease[19]. Importantly, these substitutions have not been observed in further clinical trials or surveillance studies[17], and retained full neutralizing potency from the immune serum of healthy donors. It is also notable that resistance-associated substitutions were not identified beyond 150 days post-dose (the typical length of an RSV season), suggesting that waning levels of nirsevimab (Fig. 5)[7] do not lead to an increase in the frequency of such substitutions.

This study has shown that major variant substitutions within the nirsevimab binding site have not led to a reduction in potency of nirsevimab. In fact, RSV B variants with the I206M:Q209R and Q209R substitutions exhibited increased susceptibility to nirsevimab. A potential mechanism to explain the increased susceptibility of RSV B to nirsevimab in the presence of the Q209R substitution was proposed based on molecular modeling; the Q209R substitution demonstrated improved electrostatic interaction with nirsevimab, thereby increasing the effectiveness of locking the F protein in the pre-fusion structure and preventing viral entry. This analysis highlights that not all binding site substitutions decrease the potency of monoclonal antibody prophylactics such as nirsevimab, and indeed changes in electrostatic interactions can be beneficial by increasing potency.

Limitations of this study include the small number of cases in each group that did not allow for any between-group statistical analyses. Additionally, although the primary cohort of MELODY was followed through two seasons, there were few cases of RSV past Day 150; however, this would be expected as infants are most susceptible to having an MA RSV infection during their first RSV season. With regards to qRT-PCR, limitations include differential times from symptom presentation to clinical site visits for central laboratory assessments and the possibility of higher inoculating doses of RSV being responsible for the breakthrough infections.

In summary, nirsevimab neutralized both RSV A and B subtypes obtained from participants enrolled in two pivotal clinical trials. The nirsevimab binding site was highly conserved in RSV A, and few binding site substitutions were observed in RSV B. Furthermore, microneutralization data showed that the prevalent Q209R substitution in RSV B resulted in increased susceptibility to nirsevimab and modeling provided a potential mechanism based on increased electrostatic interaction between nirsevimab and the binding site.

No resistance-associated substitutions in the F protein of RSV A isolates were identified. Although rare, resistance substitutions were identified in the F protein of RSV B in two participants in Phase 2b trial; these changes were not observed in subsequent clinical trials or molecular surveillance studies. Overall, >99% of RSV F protein sequences from the Phase 2b and MELODY trials remained susceptible to nirsevimab, and no resistance-associated substitutions were identified beyond 150 days post-dose as levels of nirsevimab waned.

## Methods
### Study designs
This was a pre-specified analysis. Full details of the methodologies used for both clinical trials have been published previously[9,11]. In brief, participants were randomly assigned 2:1 to receive one intramuscular injection of nirsevimab or placebo prior to their first RSV season (Phase 2b: 50 mg, all participants; MELODY: participants <5 kg, 50 mg; ≥5 kg, 100 mg). Participants in the Phase 2b trial were monitored for suspected LRTI or any respiratory illness requiring hospitalization over 360 days post-dose[9]. In MELODY, MA respiratory illnesses were captured with the use of standardized methods through 510 days post-dose[11]. All MELODY participants (N = 3012) were followed for 150 days post-dose; the primary cohort (N = 1490 randomized for the primary analysis) was additionally followed through 151–361 days post-dose and throughout the second season (361–510 days post-dose) to monitor for RSV disease[20]. Nasopharyngeal swabs were collected for all suspected cases of LRTI presenting for medical attention (inpatient or outpatient) and all respiratory illnesses requiring hospitalization. The per-protocol case definition for the efficacy of MA RSV LRTI required objective criteria of lower respiratory tract involvement and a clinical sign of disease severity to be present. The primary endpoint of both clinical trials was the incidence of MA RSV LRTI over 150 days post-dose; the secondary efficacy endpoint was the incidence of hospitalization due to MA RSV LRTI over the same period[9,11]. All RSV-positive infections (including LRTI and respiratory illnesses requiring hospitalization) that did not meet the stringency of the primary case definition are presented in the supplement. A summary of case definitions used in the study is included in Supplementary Table S1.

### Ethics
The trials from which these data were gathered were performed in accordance with the principles of the Declaration of Helsinki and the International Council for Harmonization Good Clinical Practice guidelines. Each site had approval from a local institutional ethics review board or ethics committee, and appropriate written informed consent was obtained for each participant. Data were collected by clinical investigators and analyzed by AstraZeneca. Neither the investigators nor the parents or guardians were aware of the trial group assignments, and all authors had access to the results of the aggregated analysis. The authors reviewed the manuscript, made the

decision to submit the manuscript for publication, and vouch for the accuracy and completeness of the data and for the fidelity of the trial to the protocol. AstraZeneca was involved in the trial design; the collection, analysis, and interpretation of the data; and the writing of the manuscript.

## Participant selection criteria

Details of the inclusion and exclusion criteria for the Phase 2b and MELODY trials have been published previously[9,11]. In brief, healthy infants ≤1 year of age upon entering their first full RSV season, with a gestational age of 29 to <35 weeks (Phase 2b) or ≥35 weeks (MELODY), were included. Infants were excluded from the study if they were eligible to receive palivizumab (according to national or local guidelines[21]), had a fever or acute illness within 7 days of randomization, or had a history of RSV disease or LRTI prior to or at the time of randomization. For inclusion in this analysis, participants were required to have NGS-evaluable RSV isolates.

Phase 2b enrollment began on November 3, 2016 with study completion on December 6, 2018. MELODY enrollment began July 23, 2019, and paused between March 15, 2020 and April 19, 2021; safety follow-up is ongoing. The data cut-off for MELODY post-pause cohort through 150 days post-dose was March 31, 2022.

## qRT-PCR and genotyping

Central laboratory testing of respiratory secretions was performed by Viracor-Eurofins (Lees Summit, MO, USA) using the 510(k) United States FDA-cleared and Conformité Européenne (European Conformity)-marked in vitro diagnostic Lyra® RSV + human metapneumovirus (hMPV) quantitative reverse transcriptase polymerase chain reaction (qRT-PCR) assay (Quidel Corporation, San Diego, CA, USA). All RSV isolates were subtyped and evaluated for genotypic and phenotypic resistance to nirsevimab and palivizumab. To characterize isolates as RSV subtype A or B, the second hypervariable region of the attachment protein G gene was amplified and sequenced using Sanger sequencing; details of the primer sequences used can be found in Supplementary Table S13. Sequencing results were compared with contemporary RSV A and B reference strains for subtype classification. Sanger sequencing of the attachment protein G was validated through measurements of analytical sensitivity (limit of detection), precision, accuracy, and stability.

The full-length RSV F gene was amplified and sequenced using NGS assays for determination of consensus polymorphism(s); major variants were defined as having ≥25% MAF, while minor variant(s) were defined as RSV A ≥ 4% to <25% MAF or RSV B ≥ 5% to <25% MAF. NGS assays (Eurofins-Viracor, MO, USA) were validated by measurements of analytical sensitivity, accuracy (i.e., detection of mutation in a mixture of sequences), and precision. RSV A and RSV B F gene sequences in FASTA format were translated to amino acid sequences and aligned against year 2013 Netherlands RSV A/13-5275 or Netherlands RSV B/13-1273 reference sequences and amino acid substitutions in the full-length RSV F protein (AA 1-574), including the nirsevimab binding site (AA 62-69 and AA 196–212[8]) and the palivizumab binding site (AA 262–275[22]) were reported in their observed context. Clinical isolates RSV A-NLD-13-005275 (GenBank® accession code KX858757.1) and RSV B-NLD-13-001273 (GenBank® accession code KX858756.1) were selected as references given at the start of nirsevimab clinical development. Assignment of RSV genotype was performed by phylogenetic clustering with a previously described 2014 reference database of 11 RSV A genotypes and 23 RSV B genotypes[23]. F protein sequences were close to the contemporary consensus F sequence based on data in Genbank. Weblogos (i.e. Sequence Logos) were generated in Seq2Logo-2.0 and colored based on amino acid chemistry[24].

Prevalence of all RSV F protein sequence variations were reported relative to consensus sequences of RSV A (N = 2875) and RSV B variants (N = 2800) collected worldwide between 2015 and 2021[14].

## Recombinant RSV variant engineering, microneutralization, and serum neutralization susceptibility assays

Recombinant RSV variants were engineered using reverse genetics to contain individual and co-occurring (for polymorphisms within the binding sites) amino acid substitutions that were observed in the Phase 2b and MELODY trials in the full-length RSV F protein. Briefly, amino acid substitutions were introduced into the complementary DNA (cDNA) of the F gene of full-length recombinant RSV A (A2-000; containing the E66 residue) or recombinant RSV B (B9320-000; containing the N197 residue) using molecular cloning techniques. Recombinant RSV variants were rescued by co-transfecting full-length antigenomic cDNA encoding the polymorphic change(s) in the F gene along with supporting plasmids encoding the ribonucleocapsid genes N, P, L, and M2-1, and plaque-purified as previously described in refs. 25,26.

Microneutralization assays were conducted by Viroclinics Biosciences BV (Rotterdam, Netherlands) in technical triplicates by serially diluting nirsevimab or palivizumab in a 96-well plate and incubating with recombinant RSV reference strain or test variant at an input of 50–1700 tissue culture infectious dose 50 per well for 1 h at 37 °C prior to the addition of Hep-2 cells (Hep-2 CCL-23, ATCC, Gaithersburg, MD, USA). Following incubation for 5 days at 37 °C, infected cells were fixed in 80% acetone and stained with a mouse anti-RSV F primary (clone 133-1H; Millipore, Cat. No. MAB858-1) and peroxidase-conjugated goat anti-mouse secondary antibody (Life Technologies, Cat. No. A16072). Tetramethylbenzidine substrate was added to the wells and RSV F expression was determined by measuring absorbance at 450 nm using a Biotek Synergy H1M plate reader. The assay was validated through the characterization of intermediate precision and repeatability. The $IC_{50}$ values were determined by fitting a four-parameter logistics model to each replicate with corrections applied based on the negative and positive controls. Geometric mean $IC_{50}$ values for each variant and reference strain were then used to calculate the fold change in variant susceptibility compared with the respective reference strain.

Serum neutralization assays were performed in technical duplicates using the microneutralization protocol described above with minimal modification. Briefly, serum from 17 donors were heat inactivated at 56 °C for 30 min prior to serial dilution and incubation with recombinant RSV test variants. Following infection on Hep-2 cells and RSV F staining, the half-maximal neutralization titer ($NT_{50}$) values were determined by fitting a four-parameter logistics model using GraphPad Prism 9.4.0. A paired statistical analysis of the corresponding recombinant viruses was performed using a two-tailed $t$-test.

As the Hep-2 cell line is commonly misidentified, cells were authenticated using PCR to identify the presence of the following markers: amelogenin, CSF1PO, D13S317, D16S539, D5S818, TH01, TPOX, and vWA.

## Molecular modeling

The crystal structure of RSV F B9320 bound to nirsevimab was downloaded from Protein Data Bank[27] (PDB ID: 5UDD). A monomeric F structure bound to nirsevimab was extracted from the complex and optimized for in silico mutagenesis in Molecular Operating Environment (MOE) 2020.09 using the default structure preparation parameters for protonation with Protonate3D, deletion of water related to crystallization, and a refinement minimization to a root mean squared gradient of 0.1 kcal/mol/A². RSV B F substitutions were introduced using the mutate function within the Protein Builder utility of MOE. The mutagenized structure was visualized and compared with the respective wild-type RSV F B9320 (PDB ID: 5UDD) and RSV F A2 (PDB ID: 5UDC) bound to nirsevimab in PyMOL version 2.2.2 (Schrodinger LLC.). Analyses of RSV conformational changes were also conducted in PyMOL. Crystal structures of RSV F B9320 bound to nirsevimab (PDB ID: 5UDD)[8], RSV F B9320 unbound (PDB ID: 5UDE)[8], RSV F A2 bound to nirsevimab (PDB ID: 5UDC)[8], and RSV F A2 unbound (PDB ID: 4MMU)[28] were downloaded from PDB. RSV F B9320 structures were aligned in

PyMOL, allowing for the quantification of conformational changes at the nirsevimab interface. The root mean squared distance (RMSD) between a residue in its bound state and its unbound state was measured in PyMOL using the measurement wizard to calculate the distance of a Cα atom of a given residue in its bound state to the same atom in its unbound state. Structural alignment and measurements were also conducted using the unbound and antibody-bound RSV F A2 structures. For both comparisons, the Cα RMSD was calculated for site Ø residues 62–74 and 197–212. RMSD measurements were plotted for each subtype using GraphPad Prism version 9.4.1.

## Pharmacokinetic analysis

Blood samples for determination of nirsevimab serum concentrations were collected pre-dose (at screening or Day 1) and 30 (MELODY only), 90 (Phase 2b only), 150, and 360 days post-dose and at the time of a MA LRTI. In MELODY, an additional blood sample was collected 7 days post-dose in Japan, while in Europe, blood was collected 15 instead of 30 days post-dose. Nirsevimab serum concentrations were analyzed as described previously in ref. 29. R version 4.0.4 was used for graphical analysis and statistical summary[30].

## Statistical analysis

A Shapiro–Wilk test was used to determine the normality of CT values from the Lyra RSV+hMPV qRT-PCR assay. Given the non-normality of CT data, a Mann–Whitney statistical test was used for statistical comparisons across groups and across studies.

## Reporting summary

Further information on research design is available in the Nature Portfolio Reporting Summary linked to this article.

## Data availability

Crystal structure data are available from the Research Collaboratory for Structural Bioinformatics Protein Data Bank[27] as follows: RSV F B9320 unbound (PDB ID: 5UDE) RSV F B9320 bound to nirsevimab (PDB ID: 5UDD) RSV F A2 unbound (PDB ID: 4MMU) RSV F A2 bound to nirsevimab (PDB ID: 5UDC) Clinical isolates selected as references at the start of nirsevimab clinical development were available from GenBank® (RSV A-NLD-13-005275: accession code KX858757.1; RSV B-NLD-13-001273: accession code KX858756.1). The raw DNA sequence data generated in this study have been deposited in the GenBank® database under accession code PRJNA989584. Trial data are subject to controlled access to ensure commitment to the Responsible Data-Sharing Principles as established by EFPIA (European Federation of Pharmaceutical Industries and Associations) and PhRMA (Pharmaceutical Research and Manufacturers of America) and guided by the Declaration of Helsinki. Any restrictions are related to ensuring the fulfillment of legal and ethical obligations to protect patients when using patient data to advance medical research. Data underlying the findings described in this manuscript will be made available within timelines required by country laws and may be obtained in accordance with AstraZeneca's data-sharing policy described at https://astrazenecagrouptrials.pharmacm.com/ST/Submission/Disclosure. Data for studies directly listed on Vivli can be requested through Vivli at www.vivli.org. Data for studies not listed on Vivli could be requested through Vivli at https://vivli.org/members/enquiries-about-studies-not-listed-on-the-vivli-platform/. AstraZeneca Vivli member page is also available, outlining further details: https://vivli.org/ourmember/astrazeneca/.

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

## Acknowledgements

The authors thank the study participants, their families, and investigators involved in the clinical trials. Medical writing support, under the direction of the authors, was provided by Richard Knight, Ph.D., and Jane Davies of CMC Connect, a division of IPG Health Medical Communications, in accordance with Good Publication Practice (GPP 2022) guidelines (Ann Intern Med 2022; 175(9):1298–1304) and was funded by AstraZeneca. Nirsevimab is being developed and commercialized in partnership between AstraZeneca and Sanofi.

## Author contributions

Study design: D.W., E.J.K., and T.V. Data analysis: B.A., K.M.T., A.A.A., D.W., J.D.G., A.K., B.S., and E.J.K. B.A., K.M.T., A.A.A., D.W., M.E.A., R.D., J.B.D., J.D.G., H.J., A.K., A.L., S.A.M., V.S.M., E.A.F.S., B.S., S.D.S., A.M.S., D.E.T., U.W.H., E.J.K., and T.V. were involved in the interpretation of data along with writing and revising the manuscript critically. B.A., K.M.T., A.A.A., D.W., M.E.A., R.D., J.B.D., J.D.G., H.J., A.K., A.L., S.A.M., V.S.M., E.A.F.S., B.S., S.D.S., A.M.S., D.E.T., U.W.H., E.J.K., and T.V. provided final approval of the version to be published and agree to be accountable for all aspects of the work.

## Competing interests

R.D. has received grants from AstraZeneca, Merck, and Pfizer; consulting fees from Merck and Pfizer; and honoraria from GSK, Merck, Pfizer, and Sanofi Pasteur. J.B.D. has received consulting fees from Sanofi; payment or honoraria from Sanofi; and has participated in data safety monitoring boards or advisory boards for AstraZeneca. S.A.M. has received grants or contracts from BMGF, GSK, Minervax, Pfizer, and the South African Medical Research Council; payments or honoraria from BMGF; and has participated in data safety monitoring boards or advisory boards for CAPRISA and PATH. E.A.S. has received grants or contracts from AstraZeneca, Johnson and Johnson, Merck, Pfizer, and Roche; consulting fees from Adiago Therapeutics, Cidara Therapeutics, Merck, Nuance Pharmaceuticals, Pfizer, and Sanofi; payment or honoraria from AstraZeneca and Pfizer; support for meeting attendance and/or travel from AstraZeneca; and has participated in data safety monitoring boards or advisory boards for Abbvie, Bill and Melinda Gates Foundation, and GSK. B.A., K.M.T., A.A.A., D.W., J.D.G., H.J., B.S., A.K., A.M.S., V.S.M., A.L., U.W.H., E.J.K., and T.V. are current employees of AstraZeneca and may hold stock or stock options. S.D.S., D.E.T., and M.E.A. are former employees of AstraZeneca.
