## [Peer Review File · Nature Communications]

Molecular and phenotypic characteristics of RSV infections in infants during two nirsevimab randomized clinical trialsReviewers' Comments:

Reviewer #1:

Remarks to the Author:

RSV infections are a major medical burden. Since 1198 monthly palivizumab administration has been used to protect high risk infants from severe RSV disease. Nirsevimab is a fully human monoclonal antibody with a marked higher neutralization activity as compared to palivizumab. In a recent Phase III study this antibody was shown to reduce RSV related hospitalization and RSV infections that require medical attention of late-preterm and term infants with about 80%. Due to the high potency of this antibody and its extended half-life this could be accomplished by just a single shot. As such Nirsevimab is intended to be used for the general infant population at a reasonable cost. Although the binding site of nirsevimab is highly conserved, general use of a monoclonal antibody might come with the risk of introducing virus variants that escape from this antibody and possibly also from other antibodies. For RSV A breakthrough infections were not associated with RSV variants that carry mutations at the antibody binding site. In the submitted report, Ahani and colleagues investigated the incidence, genotype and phenotype of RSV variants in breakthrough RSV infections among infants that were treated with nirsevimab or with placebo. For RSV B two viral variants were isolated that had mutations in the binding site that markedly reduced the sensitivity to nirsevimab mediated neutralization. Without doubt this investigation is of major medical relevance and highly appreciated. The results are clearly explained and underscore the conclusions of the authors. There are however some important questions that need to be addressed before publication. My major comment is that isolated RSV B variants should be studied in more detail.

In the study the authors isolated from 2 patients treated with nirsevimab two RSV B variants that have a strongly increased resistance to nirsevimab due to mutations in the antibody binding region. The first variant concerns a variant that next to the recurrent I206M:Q209R double mutation acquired I64T+K68E mutation, all of which are located in the antibody binding region. Although the I206M:Q209R mutation increases the sensitivity for nirsevimab, the I64T and K68E respectively resulted in a 496 and 283-fold reduction in neutralization sensitivity. A similar level of reduction in neutralization sensitivity was observed for recombinant viruses carrying the N208S mutation. Because of their impact, these three mutations and the I64T+K68E combination require more in depth analysis. Moreover, although rare in the study cohort, these mutations might increase in frequency when nirsevimab is used much more generally for multiple seasons. One important question that needs to be addressed is to what extent do the I64T and K68E mutations either apart or combined impact the viral replication of the recombinant RSV either WT or harboring the I206M:Q209R mutations. Especially the K68E substitution is a remarkable change. Similarly, what is the impact of the N208S mutation? This can be analyzed by comparing the viral replication kinetics of the respective recombinant RSV variants used in the study (recRSV WT vs recRSV_N208S vs recRSV_I64T vs recRSV_K68E and recRSV_I206M:Q209R vs recRSV_I206M:Q209R+I64T+K68E). A potential reduced fitness of the variants that escape nirsevimab would be reassuring. If clear differences in viral replication do occur, it would be highly informative if this is associated by a difference in fusion activity of the F protein variants. This can be tested by expressing the F protein (eg GFP or splitGFP) variants in mammalian cells and monitor syncytia formation via an reporter (eg co-expression of GFP or splitGFP co-expression,..).

As the binding region of nirsevimab is the most important site for neutralizing antibodies, the above described mutations of concern (I64T, K68E and N208S) could also impact neutralization by protective antibodies evoked by infection or preF vaccination. Therefore, these mutations should also be tested for their impact on the neutralizing activity by e.g reference human serum containing high levels of neutralizing antibodies or by post challenge or preF-vaccination animal serum.

To my knowledge no viral escape analysis using nirsevimab or the parental antibody (D25 or MEDI8897) has been reported yet. If in contrast such an analysis has been reported, did the above described mutations appear in this analysis? If no escape analysis has been reported the authors should perform an in vitro viral escape selection of recRSV in the presence of nirsevimab and as control palivizumab. Virals escape variants for palivizumab can readily be selected in vitro.

The authors should describe the design and production of the used recombinant viruses.

Please discuss the RSV variants that were investigated in the context of the findings of Zhu et al (Science transl med 2017) on the impact of naturally occurring RSV variants in the binding site region on the neutralizing activity of MEDI8897.

Please, include for clarity as supplementary table a list of all RSV F variants that have been isolated in this study + their respective frequency in the nirsevimab and placebo arms (i.e for both the nir and placebo arm: number of cases in which a specific variant has been isolated / number of participants in this arm)

Reviewer #2:

Remarks to the Author:

For its extended half-life and locking the RSV F protein in the pre-fusion conformation, Nirsevimab has been proved as a powerful monoclonal antibody to prevent RSV infection. In the research work, the authors tried to investigate nirsevimab binding site substitutions and the phenotypic characteristics of these substitutions during the Phase 2b and MELODY clinical trials.

Major concern:

1. Through the whole work, there is no result associated with preterm and term infants. Therefore, the title of the manuscript should be concise.

2. In Abstract, there were "242 RSV isolates collected during the Phase 2b and MELODY clinical trials". However, there were only 105 participants analyzed in the investigation. And no cell lines were used for RSV isolation.

The results concluded that "Frequency of infections caused by subtypes A and B was similar across and within the two trials". In fact, statistical results should be provided instead of conclusive sentences, a common problem in the manuscript, also shown in "which increased nirsevimab susceptibility". In the research work, only RSV infection participants with NGS results were subtyped, which cannot present the frequency of infections caused by subtypes A and B.

3. In Material and Method, The participant selection criteria should be special for the study instead of those published previously.

Why were the Netherlands RSV A and RSV B 2013 reference strains chosen?

The part of "RSV microneutralization susceptibility assay" should be the key part of the manuscript, which should be described in detail.

4. The organization of the contents of Results should be strengthened to make them easier to understand. Important results should be reflected, not just shown in the table or figure. The conclusions with "higher" should be based on statistical results.

5. In Discussion, "This study of infants with confirmed RSV, including LRTI and respiratory illnesses requiring hospitalization, demonstrated that nirsevimab effectively neutralized both RSV A and B infections in the Phase 2b and MELODY trials, including RSV infections that did not meet protocol defined endpoints." is not the conclusion of the research work. And the content of Discussion is too short to deep the significance of the research work.

Reviewer #3:

Remarks to the Author:

The manuscript from Ahani, Tuffy and colleagues investigates the molecular and phenotypic characteristics of RSV isolates from infants in Phase 2b and Phase III nirsevimab clinical trials. One nirsevimab-binding-site substitution was identified in RSV A (K209R, 2.1%), but this substitution did not affect nirsevimab susceptibility. Interestingly, two RSV B substitutions (I206M, Q209R) occurred with a frequency greater than 5% in the Phase 2b and MELODY trials, but these substitutions actually increase susceptibility to nirsevimab. Based on molecular modeling, the investigators provide a reasonable basis for the increased susceptibility, which is due to improved electrostatic interactions resulting from the Q209R substitution. There were, however, additional nirsevimab-binding-site substitutions found in RSV B, including L204S/S211N, I64T/K68E, and N208S. The I64T, K68E, and

N208S substitutions were all associated with substantially increased nirsevimab resistance (>200-fold IC₅₀). Unfortunately, the molecular basis for the increased resistance was not provided by the authors.

These are important studies for understanding the extent to which the use of nirsevimab influences RSV substitutions and evolution. The manuscript is succinct and well written, and the conclusions are supported by the data. My only major comment is that the authors are encouraged to provide a molecular basis for the resistance afforded by I64T, K68E, and N208S.

Other comments:

1) The structural figures in 2A are fairly small and general readers may have difficulty interpreting the location of the amino acid substitutions. The pink outline of the nirsevimab binding site is also difficult to discern.

2) The amino acid substitution N208S is not listed in the key for Figure 6. Is this an omission?

Please note, page and line numbers refer to the tracked changes version of the revised manuscript.

Reviewer comments	
Reviewer #1	
RSV infections are a major medical burden. Since 1198 monthly palivizumab administration has been used to protect high risk infants from severe RSV disease. Nirsevimab is a fully human monoclonal antibody with a marked higher neutralization activity as compared to palivizumab. In a recent Phase III study this antibody was shown to reduce RSV related hospitalization and RSV infections that require medical attention of late-preterm and term infants with about 80%. Due to the high potency of this antibody and its extended half-life this could be accomplished by just a single shot. As such Nirsevimab is intended to be used for the general infant population at a reasonable cost. Although the binding site of nirsevimab is highly conserved, general use of a monoclonal antibody might come with the risk of introducing virus variants that escape from this antibody and possibly also form other antibodies. For RSV A breakthrough infections were not associated with RSV variants that carry mutations at the antibody binding site. In the submitted report, Ahani and colleagues investigated the incidence, genotype and phenotype of RSV variants in breakthrough RSV infections among infants that were treated with nirsevimab or with placebo. For RSV B two viral variants were	Thank you.

isolated that had mutations in the binding site that markedly reduced the sensitivity to nirsevimab mediated neutralization. Without doubt this investigation is of major medical relevance and highly appreciated. The results are clearly explained and underscore the conclusions of the authors. There are however some important questions that need to be addressed before publication.	
My major comment is that isolated RSV B variants should be studied in more detail. In the study the authors isolated from 2 patients treated with nirsevimab two RSV B variants that have a strongly increased resistance to nirsevimab due to mutations in the antibody binding region. The first variant concerns a variant that next to the recurrent I206M:Q209R double mutation acquired I64T+K68E mutation, all of which are located in the antibody binding region. Although the I206M:Q209R mutation increases the sensitivity for nirsevimab, the I64T and K68E respectively resulted in a 496 and 283-fold reduction in neutralization sensitivity. A similar level of reduction in neutralization sensitivity was observed for recombinant viruses carrying the N208S mutation. Because of their impact, these three mutations and the I64T+K68E combination require more in depth analysis	We acknowledge the importance of investigating the impact of the observed substitutions on viral replication. However, these explorations are extensive in scale and are beyond the scope of the current manuscript. We are planning to perform analyses similar to those proposed, but these findings will be captured in a separate manuscript. Notably, fitness of the N208S substitutions was previously described (Zhu et al J Infect Dis 2018; 218: 572-580). Resistance-associated substitutions have been rare in surveillance studies and clinical studies (<1%, in both); to date none of the 3 identified substitutions resulting in decreased susceptibility to nirsevimab have circulated at a high enough frequency to be detected in global molecular surveillance studies (Wilkins et al, Lancet Infect Dis, 2023; 10.1016/s1473-3099(23)00062-2).
Moreover, although rare in the study cohort, these mutations might increase in frequency when nirsevimab is used much more generally for multiple seasons. One important question that needs to be addressed is to what extent do the I64T and K68E mutations	The effect that widespread use of nirsevimab will have on the emergence of resistant variants remains unknown. However, we anticipate that the selective pressure applied by nirsevimab usage will be limited by both the overall conservation of antigenic site Ø (limiting the pool from which to

either apart or combined impact the viral replication of the recombinant RSV either WT or harboring the I206M:Q209R mutations. Especially the K68E substitution is a remarkable change.	be selected) and the infant population (infants <1 year have a large burden of disease but are not responsible for the majority of RSV transmission). Additionally, while antigenic site Ø is the major target for neutralizing antibodies on Pre-F, it is not the only site. Importantly, recently published data show that nirsevimab provides protection from RSV disease without inhibiting an adaptive immune response (Wilkins et al. Nature Med 2023 10.1038/s41591-023-02316-5), which may additionally limit sustained viral replication and reinfection. These findings have been added to the introduction on page 4, lines 71-73 as follows: Notably, recent data demonstrate that nirsevimab protects from RSV disease without inhibiting an adaptive immune response¹².
Similarly, what is the impact of the N208S mutation? This can be analyzed by comparing the viral replication kinetics of the respective recombinant RSV variants used in the study (recRSV WT vs recRSV_N208S vs recRSV_I64T vs recRSV_K68E and recRSV_I206M:Q209R vs recRSV_I206M:Q209R+I64T+K68E). A potential reduced fitness of the variants that escape nirsevimab would be reassuring. If clear differences in viral replication do occur, it would be highly informative if this is associated by a difference in fusion activity of the F protein variants. This can be tested by expressing the F protein (eg GFP or splitGFP) variants in	A viral escape analysis for nirsevimab has previously been reported in Zhu et al. (J Infect Dis, 2018; 218: 572–580). Of the described substitutions, only N208S was presented within the viral escape analysis. However, in vitro studies did not reveal enhanced growth kinetics of the N208S variant versus the parental reference B9320. A substitution at amino acid position 68 was also described; however, it was a K68N substitution rather than the K68E observed in the current study. This variant also displayed similar growth kinetics to the parent virus. These results have now been described in the discussion, pages 14, lines 309-319 as follows:

mammalian cells and monitor syncytia formation via an reporter (eg co-expression of GFP or splitGFP co-expression,..).	Of the isolates identified from two participants in the Phase 2b trial with resistance-associated substitutions (I64T, N208S, K68E), only N208S was identified in a viral escape analysis of nirsevimab²⁴. However, when compared with the reference, the analysis did not demonstrate enhanced growth kinetics of the N208S variant in cell culture. Additionally, although a substitution at amino acid position 68 was described, it was a K68N substitution rather than the K68E substitution observed in the current study; this variant also displayed similar growth kinetics to the reference virus. Notably, these RSV variants were isolated following what was later determined to be suboptimal dosing; both participants had nirsevimab serum levels within the range of those in the weight-banded dosing regimen and also had similar viral titers to those of other nirsevimab recipients infected with RSV.
As the binding region of nirsevimab is the most important site for neutralizing antibodies, the above described mutations of concern (I64T, K68E and N208S) could also impact neutralization by protective antibodies evoked by infection or preF vaccination. Therefore, these mutations should also be tested for their impact on the neutralizing activity by e.g reference human serum containing high levels of neutralizing antibodies or by post challenge or preF-vaccination animal serum. To my knowledge no viral escape analysis using nirsevimab or the parental antibody (D25 or MEDI8897) has been reported yet. If in contrast such an analysis has been reported, did the above	Thank you for this helpful suggestion to improve our manuscript. As recommended by the reviewer, we have evaluated the nAb potency of nirsevimab against recombinant viruses expressing resistance associated substitutions in 17 serum samples derived from healthy individuals obtained prior to the RSV season). Paired analyses of matched serum samples in reference viruses compared with the resistance-associated substitutions showed that these variants were neutralized by the antibody response present in these healthy individuals, with similar antibody titers. These analyses are now described in the manuscript as follows: Page 8, lines 161-167:

described mutations appear in this analysis? If no escape analysis has been reported the authors should perform an in vitro viral escape selection of recRSV in the presence of nirsevimab and as control palivizumab. Virals escape variants for palivizumab can readily be selected in vitro.

The authors should describe the design and production of the used recombinant viruses.

Serum neutralization assays were performed in technical duplicates using the microneutralization protocol described above with minimal modification. Briefly, serum from 17 donors were heat inactivated at 56°C for 30 minutes prior to serial dilution and incubation with recombinant RSV test variants. Following infection on Hep-2 cells and RSV F staining, the half maximal neutralization titer (NT50) values were determined by fitting a four-parameter logistics model using GraphPad Prism 9.4.0. A paired statistical analysis of the corresponding recombinant viruses was performed using a two-tailed t-test.

Page 12, lines 253-258:

Given the importance of antigenic site Ø as a target of neutralizing antibodies in the immune response to RSV, resistance-associated substitutions identified in clinical studies were evaluated for their ability to be neutralized by serum from 17 healthy donors (Supplementary Figure S2). Neutralization potency was found to be either similar or improved for both N208S and I64T:K68E substitutions compared with the reference virus.

Further detail on the generation of the recombinant variants have been added to the Methods section page 7, lines 138-146 as follows.

Recombinant RSV variants were engineered using reverse genetics to contain individual and co-occurring (for polymorphisms within the

	binding sites) amino acid substitutions that were observed in the Phase 2b and MELODY trials in the full-length RSV F protein. Briefly, amino acid substitutions were introduced into the complementary DNA (cDNA) of the F gene of full-length recombinant RSV A (A2-000; containing the E66 residue) or recombinant RSV B (B9320-000; containing the N197 residue) using molecular cloning techniques. Recombinant RSV variants were rescued by co-transfecting full-length antigenomic cDNA encoding the polymorphic change(s) in the F gene along with supporting plasmids encoding the ribonucleocapsid genes N, P, L and M2-1, and plaque-purified as previously described^{17,18}. Please see the prior response to the viral escape analysis publication.
Please discuss the RSV variants that were investigated in the context of the findings of Zhu et al (Science transl med 2017) on the impact of naturally occurring RSV variants in the binding site region on the neutralizing activity of MEDI8897.	Zhu et al (J Infect Dis 2018; 218: 572-580) utilized available sequencing data up to 2014 to identify polymorphisms within the nirsevimab binding site. Their analysis found only three RSV B substitutions (K65Q, K65T, K65Q/S211N) with reduced susceptibility to nirsevimab, all of which occurred at low frequencies (1.3, 0.3 and 0.8% respectively). A larger and more contemporary molecular surveillance study reported in Wilkins et al (Lancet Infect Dis 2023; 10.1016/s1473-3099(23)00062-2) extensively focuses on naturally occurring nirsevimab binding site substitutions identified between 2015-2021. Although Wilkins et al do not report RSV B K65Q or K65T substitutions (as they were not observed between 2015-2021), they do report S211N, I206M, and Q209R as the only substitutions that occurred at frequency >1% in the nirsevimab binding site (1.1%,

68.9%, and 68.3%, respectively). Details of the Wilkins reference are included on page 14, lines 307-309, as follows:

Prospective molecular surveillance studies have also shown that the nirsevimab binding site is highly conserved among circulating RSV strains (>98% since 2016) and resistance to nirsevimab has been rare and has not increased in geo-temporal frequency¹⁶.

Importantly all of these substitutions remain susceptible to nirsevimab, as discussed on pages 14-15, lines 309-322:

Of the isolates identified from two participants in the Phase 2b trial with resistance-associated substitutions (I64T, N208S, K68E), only N208S was identified in a viral escape analysis of nirsevimab²⁴. However, when compared with the reference, the analysis did not demonstrate enhanced growth kinetics of the N208S variant in cell culture. Additionally, although a substitution at amino acid position 68 was described, it was a K68N substitution rather than the K68E substitution observed in the current study; this variant also displayed similar growth kinetics to the reference virus. Notably, these RSV variants were isolated following what was later determined to be suboptimal dosing; both participants had nirsevimab serum levels within the range of those in the weight-banded dosing regimen and also had similar viral titers to those of other nirsevimab recipients infected with RSV. These data align with recent

	reports that demonstrate no humoral antibody threshold is high enough to prevent mild RSV disease²⁵. Importantly, these substitutions have not been observed in further clinical trials or surveillance studies²³, and retain full neutralizing potency from immune serum from healthy donors.
Please, include for clarity as supplementary table a list of all RSV F variants that have been isolated in this study + their respective frequency in the nirsevimab and placebo arms (i.e for both the nir and placebo arm: number of cases in which a specific variant has been isolated / number of participants in this arm)	These data are included in Supplementary Tables 2-6 which are broken out by the corresponding protocol-defined endpoints.
Reviewer #2	
For its extended half-life and locking the RSV F protein in the pre-fusion conformation, Nirsevimab has been proved as a powerful monoclonal antibody to prevent RSV infection. In the research work, the authors tried to investigate nirsevimab binding site substitutions and the phenotypic characteristics of these substitutions during the Phase 2b and MELODY clinical trials.	
Major concern: 1. Through the whole work, there is no result associated with preterm and term infants. Therefore, the title of the manuscript should be concise	Many thanks for your observation; the inclusion of these terms was in relation to the age of the infants enrolled in the Phase 2b and MELODY trials. For clarity, ‘Preterm’ and ‘term’ have now been deleted from the manuscript title
2. In Abstract, there were “242 RSV isolates collected during the Phase 2b and MELODY clinical trials”. However, there were only 105 participants analyzed in the investigation. And no cell lines were used for RSV isolation.	The aim of this manuscript is to characterize RSV infections from two pivotal efficacy studies of nirsevimab both genotypically and phenotypically. To that end, RSV sequences from 243 study participants were evaluated genotypically. Only those substitutions which contained

The results concluded that “Frequency of infections caused by subtypes A and B was similar across and within the two trials”. In fact, statistical results should be provided instead of conclusive sentences, a common problem in the manuscript, also shown in “which increased nirsevimab susceptibility”. In the research work, only RSV infection participants with NGS results were subtyped, which cannot present the frequency of infections caused by subtypes A and B.

amino acid substitutions as compared to RSV A and B reference sequences were phenotypically evaluated in an in vitro microneutralization assay.

High-level results for both the Phase 2b and MELODY studies have previously been reported (Griffin et al. *N Engl J Med* 2020; 383: 415-425; Hammitt et al. *N Engl J Med* 2022; 386: 837-846; Muller et al. *N Engl J Med* 2023; 10.1056/NEJMc2214773). Given the alternating dominance of RSV A and B subtypes within RSV seasons (Wilkins et al. *Lancet Infect Dis* 2023; 10.1016/s1473-3099(23)00062-2) (and clinical trials), it was not pre-specified to perform statistical analyses on frequency of RSV subtypes within or between trials. Importantly, previous studies have demonstrated that nirsevimab is efficacious against both RSV A and RSV B. (Griffin et al. *N Engl J Med* 2020; 383: 415-425; Hammitt et al. *N Engl J Med* 2022; 386: 837-846; Muller et al. *N Engl J Med* 2023; 10.1056/NEJMc2214773; Simões et al. *Lancet Child Adolesc Health* 2023; 7: 180-189).

Although statistical analyses have been performed where possible (i.e. the analysis of viral load in breakthrough infections), given the nature of the analysis, namely small numbers of substitutions and the limited number of breakthrough infections in the nirsevimab arm, it has not been possible to provide fully comprehensive statistical analyses. However, we have added details of the statistical comparison between cases of RSV A and

	RSV B for infants with MA RSV LRTI and RSV LRTI requiring hospitalization at different timepoints to the footnote of Figure 1 as follows: Cases shown are those with evaluable next-generation sequencing data. Two-sided Fisher's exact tests comparing cases of RSV A and RSV B found no statistical difference by endpoint or timepoint. LRTI, lower respiratory tract infection; MA, medically attended; RSV, respiratory syncytial virus. Importantly, all RSV infections assessed by the central laboratory were subtyped, regardless of the availability of NGS sequencing and have been described previously (Griffin et al. N Engl J Med 2020; 383: 415-425; Hammitt et al. N Engl J Med 2022; 386: 837-846; Muller et al. N Engl J Med 2023; 10.1056/NEJMc2214773; Simões et al. Lancet Child Adolesc Health 2023; 7: 180-189). This manuscript builds on these publications to comprehensively characterize the specific RSV polymorphisms identified in these two pivotal studies and the impact (or lack thereof) of these substitutions on nirsevimab neutralization potency.
3. In Material and Method, The participant selection criteria should be special for the study instead of those published previously.	As the participants were from the Phase 2b and MELODY trials, the overall inclusion/exclusion criteria for these trials remain relevant for this analysis. For specific inclusion in this analysis, participants were required to have next generation sequencing- (NGS-) evaluable RSV isolates. The

Why were the Netherlands RSV A and RSV B 2013 reference strains chosen?

'Participant selection criteria' section (page 6, line 109-111) has been revised as follows:

Details of the inclusion and exclusion criteria for the Phase 2b and MELODY trials have been published previously^{9,11}. To be included in this analysis, participants were required to have next generation sequencing- (NGS-) evaluable RSV isolates.

Clinical isolates RSV A-NLD-13-005275 (GenBank® accession no. KX858757) and RSV B-NLD-13-001273 (GenBank® accession no. KX858756) were selected as references given at the start of nirsevimab clinical development, F protein sequences were close to the contemporary consensus F sequence based on data in Genbank. These details have been clarified on page 7, lines 126-132, as follows:

RSV A and RSV B F gene sequences in FASTA format were translated to amino acid sequences and aligned against the Netherlands RSV A and RSV B 2013 reference strains respectively to assess amino acid variation. Clinical isolates RSV A-NLD-13-005275 (GenBank® accession no. KX858757) and RSV B-NLD-13-001273 (GenBank® accession no. KX858756) were selected as references given at the start of nirsevimab clinical development, F protein sequences were close to the contemporary consensus F sequence based on data in Genbank.

The part of “RSV microneutralization susceptibility assay” should be the key part of the manuscript, which should be described in detail.

Additional details on the microneutralization susceptibility assay previously included in the supplement have been moved to page 7-8, lines 147-160, as follows:

Microneutralization assays were conducted by Viroclinics Biosciences BV (Rotterdam, Netherlands) in technical triplicates by serially diluting nirsevimab or palivizumab in a 96-well plate and incubating with recombinant RSV reference strain or test variant at an input of 50-1700 tissue culture infectious dose 50 per well for 1 hour at 37°C prior to the addition of HEp-2 cells. Following incubation for 5 days at 37°C, infected cells were fixed in 80% acetone and stained with a mouse anti-RSV F primary (Millipore, Cat. No. MAB858-1) and peroxidase conjugated goat anti-mouse secondary antibody (Life Technologies, Cat. No. A16072). Tetramethylbenzidine substrate was added to the wells and RSV F expression was determined by measuring absorbance at 450 nm using a Biotek Synergy HIM plate reader. The assay was validated through characterization of intermediate precision and repeatability. The half maximal inhibitory concentration (IC_{50}) values were determined by fitting a four-parameter logistics model to each replicate with corrections applied based on the negative and positive controls. Geometric mean IC_{50} values for each variant and reference strain were then used to calculate the fold change in variant susceptibility compared with the respective reference strain.

4. The organization of the contents of Results should be strengthened to make them easier to understand. Important results should be reflected, not just shown in the table or figure. The conclusions with “higher” should be based on statistical results.

The authors recognize that supplementary figures can be distracting but note that these analyses are important for fully describing and interrogating the cause for MA RSV LRTI infection, including in the nirsevimab group. Please note that the main figures in the results flow as follows:

1. High-level overview of number of cases of RSV, by study and subtype
2. Genotypic analyses of the RSV infections described in Figure 1
3. Analysis of the nirsevimab binding site from the genotypic analyses in Figure 2
4. Phenotypic analysis of all individual substitutions identified in the study, including those seen in Figures 2/3
5. Deep-dive into the two infants who had resistant substitutions identified in Figure 4
6. Deep-dive into the highly prevalent nirsevimab binding site substitution that had higher potency than the reference amino acid
7. Molecular modeling of the binding site substitution that is characterized in Figure 6 that shows the mechanism for the increase in potency

This organization is logical to the authors. We have added further description to the results section; however, as per Nature editorial style, we have left interpretation of the data to the Discussion.

5. In Discussion, “This study of infants with confirmed RSV, including LRTI and respiratory illnesses requiring hospitalization, demonstrated that nirsevimab effectively neutralized both RSV A and B infections in the Phase 2b and MELODY trials, including RSV infections that did not meet protocol defined endpoints.” is not the conclusion of the research work. And the content of Discussion is too short to deep the significance of the research work.	Thank you for the opportunity to clarify the key messages of our manuscript and further expand the discussion. A tracked changes copy of the manuscript and its updated discussion are included with this response.
Reviewer #3	
The manuscript from Ahani, Tuffy and colleagues investigates the molecular and phenotypic characteristics of RSV isolates from infants in Phase 2b and Phase III nirsevimab clinical trials. One nirsevimab-binding-site substitution was identified in RSV A (K209R, 2.1%), but this substitution did not affect nirsevimab susceptibility. Interestingly, two RSV B substitutions (I206M, Q209R) occurred with a frequency greater than 5% in the Phase 2b and MELODY trials, but these substitutions actually increase susceptibility to nirsevimab. Based on molecular modeling, the investigators provide a reasonable basis for the increased susceptibility, which is due to improved electrostatic interactions resulting from the Q209R substitution. There were, however, additional nirsevimab-binding-site substitutions found in RSV B, including L204S/S211N, I64T/K68E, and N208S. The I64T, K68E, and N208S substitutions were all associated with substantially	Thank you for the thoughtful review and the opportunity to improve the manuscript. In regards, to the molecular basis of the I64T:K68E and N208S substitutions, we have now performed molecular modeling of these substitutions in RSV F and the analysis is included on page 13, lines 281-288 as follows: A subsequent investigation into the RSV B I64T:K68E and N208S substitutions that were associated with nirsevimab resistance was also performed. I64, K68, and N208 have all been reported to interact with nirsevimab, and the resulting K68E and N208S in silico substitutions led to the loss and reduction of polar contacts, respectively (Supplementary Figure 5). I64 is near the heavy chain CDR3 of nirsevimab and forms a close contact with V99 (<4 Å). However, the I64T in silico substitution did not impact the distance between these residues (Supplementary Figure 5). Instead, the decrease in nirsevimab potency due to this

increased nirsevimab resistance (>200-fold IC50). Unfortunately, the molecular basis for the increased resistance was not provided by the authors. These are important studies for understanding the extent to which the use of nirsevimab influences RSV substitutions and evolution. The manuscript is succinct and well written, and the conclusions are supported by the data. My only major comment is that the authors are encouraged to provide a molecular basis for the resistance afforded by I64T, K68E, and N208S.	substitution may stem from introducing a polar side chain into the site Ø-nirsevimab interface, thus disrupting a patch of hydrophobic interactions.
Other comments: 1) The structural figures in 2A are fairly small and general readers may have difficulty interpreting the location of the amino acid substitutions. The pink outline of the nirsevimab binding site is also difficult to discern.	The figure has been revised to make the greys paler so that the colors are more prominent/clear. Regarding the size of the figures, this will depend on the final layout, which is beyond the control of the authors.
2) The amino acid substitution N208S is not listed in the key for Figure 6. Is this an omission?	N208S was observed as an individual polymorphism and did not co-occur with I206M:Q209R. Therefore, it was not included in the analysis for Figure 6 which only investigated substitutions that co-occurred with I206M:Q209R.

Reviewers' Comments:

Reviewer #1:

Remarks to the Author:

The authors provided clear responses included clarifications in the text and performed and included informative neutralization experiments as requested. If also the comments by the other reviewers are adequately addressed I consider the manuscript suited for publication.

Reviewer #2:

Remarks to the Author:

The manuscript has been carefully revised based on the reviewers' comments.

Minor concerns:

1. HEp-2 or Hep-2 should be unified.
2. The conclusions that "The frequency of infections caused by subtypes A and B was similar" and "nirsevimab effectively neutralized both RSV A and B infections in the Phase 2b and MELODY trials" should be supported by statistical analysis data.
3. In the manuscript, qRT-PCR was used to confirm RSV infection rather than RSV titer. Therefore, the descriptions that "A Shapiro-Wilk test was used to determine the normality of cycle threshold (CT) values from the Lyra RSV+hMPV qRT-PCR assay. Given the non-normality of CT data, a Mann-Whitney statistical test was used for statistical comparisons across groups and across studies" and "At the time of symptomatic illness, RSV titers were similar in participants with breakthrough infections, regardless of whether the virus had resistance associated substitutions (Supplementary Figure S3). Median CT values were numerically higher (corresponding to lower viral load) among nirsevimab recipients compared with placebo recipients in the Phase 2b trial, but similar in the MELODY trial, with no statistical significance shown in either trial" were very strange.

Reviewer #3:

Remarks to the Author:

The authors have adequately addressed my prior comments in the revised manuscript.

Please note, page and line numbers refer to the tracked changes version of the revised manuscript.

Reviewer comments	Response
Reviewer 1	
The authors provided clear responses included clarifications in the text and performed and included informative neutralization experiments as requested. If also the comments by the other reviewers are adequately addressed I consider the manuscript suited for publication.	Thank you.
Reviewer 2	
The manuscript has been carefully revised based on the reviewers' comments. Minor concerns:	Thank you
1. HEp-2 or Hep-2 should be unified.	This has been unified to Hep-2 throughout the manuscript.
2. The conclusions that “The frequency of infections caused by subtypes A and B was similar” and “nirsevimab effectively neutralized both RSV A and B infections in the Phase 2b and MELODY trials” should be supported by statistical analysis data.	With regards to “nirsevimab effectively neutralized both RSV A and B infections in the Phase 2b and MELODY trials”, the analysis was not powered to statistically confirm that every rare variant that occurred in these studies was neutralized. We have, however, revised this text here to provide additional clarity (page 9, lines 180-186): This study of infants with confirmed RSV, including LRTI and respiratory illnesses requiring hospitalization, demonstrated that nirsevimab effectively neutralized both RSV A and B infections among participants enrolled in the Phase 2b and MELODY trials, regardless of the presence of naturally occurring amino acid substitutions, and

	including RSV infections that did not meet protocol defined endpoints. Genotypic analyses showed that most substitutions occurred at a low frequency and in non-EC portions of RSV F rather than in antigenic sites, indicating that the nirsevimab binding site remains conserved and thus RSV remains susceptible to nirsevimab.
3. In the manuscript, qRT-PCR was used to confirm RSV infection rather than RSV titer. Therefore, the descriptions that “A Shapiro-Wilk test was used to determine the normality of cycle threshold (CT) values from the Lyra RSV+hMPV qRT-PCR assay. Given the non-normality of CT data, a Mann-Whitney statistical test was used for statistical comparisons across groups and across studies” and “At the time of symptomatic illness, RSV titers were similar in participants with breakthrough infections, regardless of whether the virus had resistance associated substitutions (Supplementary Figure S3). Median CT values were numerically higher (corresponding to lower viral load) among nirsevimab recipients compared with placebo recipients in the Phase 2b trial, but similar in the MELODY trial, with no statistical significance shown in either trial” were very strange.	qRT-PCR is the most commonly used diagnostic assay for detection of RSV infection in both clinical practice and for epidemiological surveillance and, in the view of the authors, is the most appropriate test for these clinical studies. Given volume limitations for a single swab in UTM, it was not possible to perform infectious viral titration after central laboratory diagnostic assay, sanger sequencing (utilized to subtype RSV), and NGS (utilized for determination of amino acid polymorphisms). While it is interesting that nirsevimab did not reduce viral load in breakthrough infections, as determined by Ct value in qRT-PCR, limitations to this analysis include differential times from symptom presentation to clinical site visit for central laboratory assessments and the possibility of higher inoculating doses of RSV being responsible for the breakthrough infections. The latter limitations of qRT-PCR have been added to the limitations section on page 11, lines 229-231, as follows: With regards to qRT-PCR, limitations include differential times from symptom presentation to clinical site visit for central laboratory

	assessments and the possibility of higher inoculating doses of RSV being responsible for the breakthrough infections.
Reviewer 3	
The authors have adequately addressed my prior comments in the revised manuscript.	Thank you